# Spatial Attention Modulates Neuronal Interactions between Simple and Complex Cells in V1

**DOI:** 10.3390/ijms24098229

**Published:** 2023-05-04

**Authors:** Zhiyan Zheng, Qiyi Hu, Xiangdong Bu, Hongru Jiang, Xiaohong Sui, Liming Li, Xinyu Chai, Yao Chen

**Affiliations:** School of Biomedical Engineering, Shanghai Jiao Tong University, Shanghai 200240, China

**Keywords:** spatial attention, neuronal interaction, heterogeneous visual input, Granger causality, primary visual cortex

## Abstract

Visual perception is profoundly modulated by spatial attention, which can selectively prioritize goal-related information. Previous studies found spatial attention facilitated the efficacy of neuronal communication between visual cortices with hierarchical organizations. In the primary visual cortex (V1), there is also a hierarchical connection between simple (S) and complex (C) cells. We wonder whether and how spatial attention modulates neuronal communication within V1, especially for neuronal pairs with heterogeneous visual input. We simultaneously recorded the pairs’ activity from macaque monkeys when they performed a spatial-attention-involved task, then applied likelihood-based Granger causality analysis to explore attentional modulation of neuronal interactions. First, a significant attention-related decrease in Granger causality was found in S-C pairs, which primarily displayed in the S-to-C feedforward connection. Second, the interaction strength of the feedforward connection was significantly higher than that of the feedback under attend toward (AT) conditions. Although information flow did not alter as the attentional focus shifted, the strength of communications between target- and distractor-stimuli-covered neurons differed only when attending to complex cells’ receptive fields (RFs). Furthermore, pairs’ communications depended on the attentional modulation of neurons’ firing rates. Our findings demonstrate spatial attention does not induce specific information flow but rather amplifies directed communication within V1.

## 1. Introduction

A primate’s brain has a limited capacity to process all visual information, whereas attention can prioritize goal-related information selectively. Monkeys, for example, can effectively find bananas in the wild with the help of selective visual attention. In neurophysiological studies, the consequences of shifting the attentional focus on the receptive fields (RFs) of individual neurons are well understood. That is, most neurons in the visual cortex show a modest change in their responses (i.e., firing rates magnitude, firing rate variability) when spatial attention is directed toward their RFs (for reviews, see [1,2]). However, the brain relies on multiple neurons working together to recognize objects and accomplish goals. Whether and how spatial attention modulates the communications among neurons remains unknown.

Many studies have been carried out to answer the question above. They found spatial attention facilitated the efficacy of neuronal communication between visual areas with the hierarchical organization for information processing, such as lateral geniculate nucleus (LGN)—V1 [3], V1—V4 [4,5], V4—frontal eye field (FEF) [6], V1—middle temporal (MT) [7], and superior colliculus (SC)—MT [8]. Neurons in V1 can be classified as simple and complex cells, with a hierarchical functional connection between them [9,10,11]. That is, complex cells integrate convergent inputs from simple cells [12,13]. We wonder whether spatial attention also modulates neuronal communication within V1 as in those across visual areas. Hembrook-Short et al. found feedforward local circuits from simple to complex cells within V1 displayed the largest improvement with spatial attention [14]. However, as with the cross-cortical studies, they used a large grating stimulus covering both the RFs of paired neurons. Spatial attention plays the same role in both neurons under such an experimental design. Therefore, we simultaneously recorded paired single units in V1 and used likelihood-based Granger causality (GC) analysis to further explore how spatial attention influences neuronal communication when the attentional focus is shifted from one neuron’s RF to the other’s.

Granger causality (proposed by [15]) is an efficient algorithm for studying causative connections in sensorimotor, visual, and prefrontal cortical networks [4,6,16]. Given neuronal spike-train activity occurs as the point processes rather than samples of continuous processes, we applied the Granger causality analysis based on the likelihood approach [17]. This analysis distinguished excitatory effects on the neural interactions by a positive GC value and inhibitory communications by a negative GC value. Researchers have applied it to assess directional interactions among neuronal signals for both monkeys (i.e., [18]) and mice (i.e., [19]).

Our goal was to determine whether and how spatial attention modulated communications between neuronal pairs in V1 with heterogeneous visual input. We proposed that spatial attention would enhance communication within the S-C hierarchy, following the findings reported in cross-visual-area studies. Moreover, pairs’ communication would be inhibitory due to heterogeneous visual input. Therefore, we recorded the responses to heterogeneous gratings of neuronal pairs by two independent electrodes from V1 of awake monkeys while they performed a spatial-attention-involved task (Figure 1). The attentional focus could be directed to each neuron’s RF separately. We first explored whether there is an attentional enhancement of communication in V1 or solely within the S-C hierarchy. We then examined how spatial attention modulated the communications within this hierarchy, as well as the key factors that influenced the attentional modulation of hierarchical communications.

## 2. Results

To investigate how spatial attention affects interactions between two V1 neurons, we recorded single-unit activities by using a 32-electrodes micro-drive system. We simultaneously recorded the activity of two V1 neurons responding to the heterogeneous gratings while monkeys were performing a covert, top-down, spatial attention task, then were rewarded after successfully detecting the color change of the cued grating (Figure 2, left). The attentional focus was manipulated by the peripheral cue. We determined the attentional conditions based on the relative position of the attentional focus and recorded neurons’ RFs (Figure 2, upper right). Then the interactions between paired neurons were estimated by Granger causality.

We calculated detection accuracy between trials with valid and invalid cues to examine if the cue successfully directed the animals’ attentional focus. We found that when target stimuli were presented in the cued location, the accuracy increased significantly (*t* = 5.08, *p* = 3 × 10^−3^, paired *t*-test), whereas the position of the cue had no significant impact on either detection accuracy (*F* = 0.53, *p* = 0.47, rmANOVA) or reaction time (*F* = 0.02, *p* = 0.90, rmANOVA).

### 2.1. Neuronal Classification and Attentional Modulation on Firing Rates

Recorded neurons were classed as simple (S) and complex (C) cells by calculating the response linearity (F1/F0). The distributions for neurons’ response linearity are shown in Figure 3A. After classification, we paired the recorded neurons as S-S, C-C, and S-C pairs. The magnitude of attentional effects on individual neurons’ firing rate was estimated by the attentional ratio (AR = (AT_In − AT_Out)/(AT_In + AT_Out)). Figure 3B shows the AR distributions of simple and complex cells. A positive AR indicates attention-related enhancement of firing rates (Figure 3C, upper); conversely, a negative AR shows attention-related suppression (Figure 3C, down). We also multiplied the AR of each neuron to measure the attentional effects on neuronal pairs and defined it as multiplicative attentional ratios (MAR). Figure 3D shows the MAR distributions of S-S, C-C, and S-C pairs. See Materials and Methods for the detailed calculation of AR and MAR.

### 2.2. Attentional Modulation of Neuronal Interactions

We found that spatial attention significantly influenced the interactions between simple and complex cells (*F* = 4.72, *p* = 0.032, rmANOVA). Although the strength of communications between S-to-C and C-to-S was not significantly different (*F* = 2.87, *p* = 0.093, rmANOVA), they were significantly modulated by spatial attention (*F* = 6.19, *p* = 0.015, rmANOVA). As seen in Figure 4A, attending toward the RFs of pairs significantly decreased the GC value to be negative in the feedforward S-to-C direction (*t* = −2.31, *p_bonf_* = 0.020, Bonferroni post hoc test), but it had no significant effect on the C-to-S direction (*t* = −1.96, *p_bonf_* = 0.052, Bonferroni post hoc test). Furthermore, only under the AT conditions was the communication strength of the S-to-C direction significantly stronger than the C-to-S direction (*t* = −2.31, *p_bonf_* = 0.023, Bonferroni post hoc test). For S-S (Figure 4B) and C-C pairs (Figure 4C), spatial attention did not affect the communication strength and directions (all *p* > 0.05, rmANOVA).

### 2.3. The Attentional Focus Influenced S-C Pairs’ Interactions

Under the AT conditions, the attentional focus was directed to one of the neuronal pairs’ RFs by the cue (Figure 5A). We classified cued stimuli as targets and others as distractors. To investigate how the attentional focus affected neuronal communications in the AT conditions, neurons were divided into target-stimuli-covered neurons (T neurons) and distractor-stimuli-covered neurons (D neurons). We calculated GC from T neuron to D neuron (T → D), and vice versa (D → T). Figure 5B demonstrates that there was no significant difference between these two communicating directions (*F* = 1.63, *p* = 0.20, rmANOVA) and that the types of neuronal pairs had no significant effect on it (*F* = 1.54, *p* = 0.22, rmANOVA).

We further investigated whether and how the attentional focus might affect communication patterns between simple and complex cells in the AT conditions. When the attentional focus shifted from simple cells’ RFs to those of complex cells, ΔGC significantly changed from positive to negative (*F* = 5.54, *p* = 0.021, rmANOVA; Figure 5C). Moreover, the GC of T → D was significantly more negative than D → T when cued stimuli covered complex cells’ RF (*t* = −2.11, *p_bonf_* = 0.038, Bonferroni post hoc test; Figure 5C) instead of simple cells’ RFs (*t* = 1.44, *p_bonf_* = 0.16, Bonferroni post hoc test; Figure 5C).

### 2.4. Pairs’ Interactions Depend on Attentional Modulation of Firing Rates

As spatial attention exhibited either enhanced or suppressed effects on neuronal firing rates (i.e., [20]), we further investigated whether the sign of attentional modulation of each neuron’s activities influenced their interactions. When MAR was negative, the communication strength of S-to-C was significantly stronger than that of C-to-S (*t* = −2.15, *p* = 0.037, paired *t*-test; Figure 6A). The differences vanished when MAR was positive (*t* = −1.05, *p* = 0.30, paired *t*-test; Figure 6A). Furthermore, only when MAR was negative were the communicating directions between T and D neurons reversed as the attentional focus placement shifted (*F* = 4.54, *p* = 0.038, rmANOVA; Figure 6B). Only when the attentional focus was directed toward complex cells’ RFs did GC of T → D become significantly more negative than the opposite directions (*t* = −2.07, *p_bonf_* = 0.049, Bonferroni post hoc test; Figure 6B).

### 2.5. Attentional Impacts on Interactions Depending on the Layers Neurons Originated from

In our study, most of the recorded neurons were from the supra-granular layer. We observed the similar attention-modulating effects described above when pairs were placed within the supra-granular layer (Figure 7A). That is, spatial attention significantly affected the strength of S-C pairs’ communications (*F* = 5.02, *p* = 0.028, rmANOVA). In addition, it significantly modulated the communicating directions (*F* = 4.75, *p* = 0.033, rmANOVA). Moreover, attention significantly decreased the GC values of both S-to-C (*t* = −2.31, *p_bonf_* = 0.024, Bonferroni post hoc test) and C-to-S (*t* = −2.12, *p_bonf_* = 0.037, Bonferroni post hoc test) directions to be negative. S-to-C communication strength was significantly stronger than C-to-S direction only under the AT conditions (*t* = −2.24, *p_bonf_* = 0.028, Bonferroni post hoc test). When neither of the neurons came from the supra-granular layer, these spatial attentional effects disappeared (all *p* > 0.05, rm ANOVA and Bonferroni-corrected post hoc analysis; Figure 7B).

## 3. Discussion

We used the likelihood-based GC as a measure of directional communication to investigate how spatial attention affected neuronal interactions within V1. The recorded neurons with non-overlapping RFs received heterogeneous visual input. Therefore, our observation of significant attentional enhancement of inhibitory communication in the S-C hierarchy supports the hypothesis we came up with. Subsequently, we compared our findings with previous publications, elucidated the physiological implications of the results, and then pointed out the limitations and prospects for potential future explorations.

### 3.1. Comparison to Previous Neuronal Communication Studies

We observed that GC decreased to negative when spatial attention was directed into one of the neurons’ RFs between simple and complex cells. These results indicated the enhancement in competing connections between pairs with hierarchical architecture, which supported our hypothesis. Moreover, it was also consistent with previous trans-cortical [3,4,5,6] and homo-cortical studies [14,21], which demonstrated attentional-related facilitation in neuronal communications.

However, the underlying mechanism in our study may differ from previous studies. In these previous studies, neurons’ RFs were covered by a sufficiently big stimulus, which means they cooperated to recognize the target. However, neurons in our study received different visual inputs, implying that the pattern of communication was different. That is, when the attentional focus switched away from neurons’ RFs, pairs performed a similar interference function in identifying targets. In this case, the communication between them is meaningless and energy-consuming. However, when the attentional focus shifted to one neuron’s RF, the information from the other neuron became interference [20,22,23]. Based on that, spatial attention would suppress the noise to improve the efficacy of encoding sensory. Another aspect might be the synchronization of neural responses. Previous studies calculated phase-amplitude coupling [24], spike-phase coupling [25], and phase coherence [26] to investigate the effect of spatial attention on neuronal synchronization. They discovered a spatial-attention-induced reduction in neuronal synchrony in both the pre-stimulus period [26] and stimulus presentation period [24,25]. What is more, numerous studies have found gradual increases in the attentional modulation of neuronal firing rates throughout hierarchically structured areas [6,27,28,29,30]. It could also partially explain why relatively stronger effects of spatial attention were found between simple and complex cells in our experiment.

The attentional modulation of GC was prominently seen in the bottom-up directions. It was consistent with previous studies [4,14], which also discovered that feedforward interactions displayed a larger enhancement with spatial attention. Our studies provide new evidence on the hierarchical progression of attentional modulation. Furthermore, under AT conditions, we found that GC was substantially stronger in the feedforward S-to-C direction than vice versa. The results support the concept of hierarchical structure between simple and complex cells: complex cells’ RFs are generated from simple cells, indicating that they further integrate input from simple cells [12,13]. These results also shed further light on how spatial attention modulates communications within the S-C hierarchy. That is, attention specifically amplifies feedforward information flow, thus enhancing visual perception.

However, in this study, spatial attention did not affect the communication strength and directions for S-S and C-C pairs. It could be attributable to anatomical characteristics. Both S-S and C-C pairs belong to the same level of the hierarchy within V1. Hence, it is plausible that spatial attention did not have an impact on their information flow. For S-S pairs, another reason might be the lack of sufficient statistical power. Most of the neurons we recorded were from the supra-granular layer because attention predominantly modulates neuronal activities in the supra-granular layers [31,32,33,34]. However, previous studies reported that simple cells are mostly observed in layer 4 [35,36] and converge into the superficial layer to generate complex cells’ RFs [10,12,37]. Thus, in this study, we rarely recorded two simple cells simultaneously. We applied a priori power analysis by G*power (software based on hypothesis testing) to determine the sample size (α = 0.05, power = 0.8) with a medium effect size for every statistical test we used (the parameters were set by referencing [38,39]). It suggested a minimum sample size of 17 for rmANOVA and 15 for paired *t*-tests, both of which were slightly smaller than the S-S pairs’ sample size.

### 3.2. The Influence of Attentional Focus Placement on S-C Pairs

Considering target and distractor stimuli would compete with one another for better representation, we classified neurons as target-stimuli-covered and distractor-stimuli-covered neurons. We found that there were no significant information flow differences between these three types of pairs. It implies that spatial attention itself does not induce specific information flow but rather amplifies directed communication within V1. It clarifies that spatial attention acts as an amplifier of neuronal communications.

When the attentional focus switched from the RFs of simple cells to those of complex cells, the communicating directions between T and D neurons were reversed. Since simple and complex cells were competitively connected under AT conditions, this result indicated that information flowed more from simple to complex cells regardless of which neurons’ RFs were covered by attentional focus. It further supports the hierarchical hypothesis of simple and complex cells mentioned above. What is more, we discovered that it was only when the attentional focus shifted to complex cells’ RFs that the information exchange directions made a significant difference. To detect external visual information more effectively, complex cells in V1 need to integrate input from simple cells, while simple cells mainly handle input from LGN [40]. Thus, it was simple to discover in this study that complex cells received significantly more competing input from simple cells when cued stimuli covered complex cells’ RFs.

### 3.3. S-C Pairs’ Communication Depends on Their MAR

Given that attentional modulation of neuronal activity varies even with a visual brain area [20,41,42], we further investigated whether and how attention-related enhancement or suppression effects on individual neurons’ activities influence their interactions. When multiple stimuli are presented, selected and unselected stimuli compete with one another for better representation [23,43,44,45]. The V1 neuronal pairs recorded in our study simultaneously received two competing stimuli inputs when attentional focus covered one of the neurons’ RF. We discovered that the communication strength was significantly stronger in the feedforward S-to-C direction when MAR was less than zero, while these differences vanished when MAR was larger than zero. Furthermore, only when MAR was less than 0 could the similar effects of attentional focus placement be found between simple and complex cells.

This is probably because spatial attention has the same effect on both neurons’ firing rates when MAR is positive. When cued stimuli cover one neuron’s RF, neuronal pairs display an increase in one neuron’s firing rate and a decrease in the other neuron’s firing rate. Previous studies suggested that neuronal spiking activities enhance neuronal connectivity [46,47,48]. The firing rates of one neuron decrease under AT conditions, causing the interactions between the neuronal pairs to converge to zero. Therefore, we cannot observe the differences in information flow between simple and complex cells. However, spatial attention plays the opposite role in pairs’ firing rates when MAR is less than zero, causing both neurons’ firing rates to vary in the same way when the attentional focus shifts to one neuron’s RF. In such cases, the relative input from another neuron was stronger. Thus, there is a greater need for the target-stimuli-covered neuron to suppress the noise from another neuron.

### 3.4. The Layers Neurons Come from Explains Attentional Impact on S-C Pairs’ Interaction

The fact that most neurons recorded in our study came from supra-granular layers might explain why we discovered significant attentional modulation of neuronal communication between simple and complex cells. In this study, 72 out of 96 S-C pairs we recorded were placed within supra-granular layers, 20 out of 96 had one neuron from the supra-granular layers and another from the infra-granular layers, 2 out of 96 had one neuron from the granular layers and another from the infra-granular layers, and the rest were simultaneously originated from infra-granular layers. We found top-down attention had a significant impact on neuronal communication only when both neurons of S-C pairs were from the supra-granular layers. Three pieces of evidence are available to help understand it. First, a previous study found attentional modulation of neurons’ correlations was strongest when neurons came from the same layer [41]. In this study, up to seventy-seven percent of neuronal pairs recorded simultaneously were from the same layer. Second, neuronal spiking is almost exclusively synchronized in the superficial layers in both V1 and V4, while it lacks synchronization in infra-granular layers [49]. Third, neurons from V1 that project to V4 are virtually entirely found in the supra-granular layers [4,50]. These neurons are therefore more susceptible to top-down, spatial attention. Researchers also found top-down attention predominantly modulates neurons’ firing rates and correlations in the supra-granular layers [31]. Thus, the finding in S-C pairs matches well with the known anatomy characteristics of cortical interactions [33,34].

### 3.5. Other Potential Confounders That Cannot Explain Our Findings

First, we analyzed the eye position across attention conditions of 1000 ms time windows before the color change. The average deviation of eye position away from the central fixation was only about 0.11° (SD = 0.01°) in all sessions. Given that micro-saccades (also called fixational eye movements) have been reported to modulate neuronal spiking activities (i.e., [51]) and neuronal response variability (i.e., [52]), we found the direction and frequency of micro-saccades did not vary across attentional conditions (all *p* > 0.05, paired *t*-test).

Second, we chose the parameters of gratings to evoke a strong enough response in neurons and fixed these parameters in our experiment. The spatial frequency preferences for neurons in V1 ranged from 0.5 to 8 cycles/degree, while these neurons responded well at the temporal frequency up to 5.6 cycles/s and dropped off their responses at a higher temporal frequency [53]. The low-contrast stimuli result in extremely low neuronal firing rates [7,54], while gratings with a diameter of 2–3° evoke higher neuronal responses by fitting the size-tuning curves [54]. Thus, we designed the grating’s characteristics based on the typical preference of neurons in V1 to elicit a strong enough neuronal response and minimize the impact of variables other than spatial attention on neuronal responses.

Third, considering that the individual neuron’s activity would influence its interaction with other neurons, we further analyzed the possible effect of attention modulation on the individual neuron’s firing rate. We found neither the AR sign of the T neuron nor the D neuron affected the communicating directions between simple and complex cells (all *p* > 0.05, rmANOVA).

### 3.6. Limitations

There are still some limitations in our study. First, most neurons recorded in our study were obtained from supra-granular layers to ensure that the effect of attention on neuronal activities was strong enough. Follow-up studies could use multichannel linear electrodes to record neuronal activities from multiple layers simultaneously. By doing so, we can better understand the hierarchical structure of simple and complex cells in V1 and the role of attention in the early stage of visual processing. Second, although the sample size of S-S and C-C pairs exceeded the minimum number calculated by prior power analysis, it would be better if the number of these two types of pairs was closer to that of S-C pairs. Third, each electrode in the recording system we used can be controlled independently by a screw. However, compared to other electrophysiological recording systems (such as the Utah array), the channel spacing is large and the number of neurons that can be recorded in each session is limited. It may be advantageous to utilize the Utah array to simultaneously record the activities of pairs that received homogeneous or heterogeneous visual input.

## 4. Materials and Methods

### 4.1. Subjects

Two adult male rhesus macaque monkeys (*Macaca mulatta*; monkey P: 7.5 kg; monkey S: 9 kg) were included in this study. Animals were individually housed with four other monkeys on a 12 h light/dark cycle. Husbandry, surgeries, and experimental procedures were carried out according to the NIH guidelines and the Institutional Animal Care and Use Committee of Shanghai Jiao Tong University.

### 4.2. Spatial Attention Behavioral Task

During the experiment, monkeys’ eye movements were measured by eye-tracking equipment (1000 samples/s; ScleraTrak 4000, Crist Instrument Co., Hagerstown, MD, USA). Monkeys had to direct and keep their gaze at the centrally placed white spot, with a fixation window of 0.8–1.0° of visual angle. At 100 ms after monkeys fixated the central spot, a red ring (diameter: 3°) was presented peripherally for 400 ms.

After the cue vanished, four gratings occurred (temporal frequency: 2 cycles/s; spatial frequency: 0.5 cycles/degree, contrast: 90%), located equidistant from the central spot. Two stimuli covered recorded pairs’ RFs separately (attend toward condition), and the other two were located outside the RF (attend away condition). The direction of the grating that covered the recorded neuron’s RF was the preferred direction of that neuron (the direction that evoked the strongest neuronal activities). If the preferred directions of the recorded pairs were matched, the directions of two gratings outside the neuronal RFs would be the same. Otherwise, the other two gratings’ directions would be chosen at random and would differ from one another.

After a random delay, cued stimuli started to turn red. Monkeys had to make a saccade to the color-changed stimuli within 500 ms to receive a juice reward. We adjusted the red value of the target stimuli (range from 8 to 100) under the performance curves of two monkeys to maintain a steady behavioral performance (the detective accuracy was more than 70%). Appendix A shows the performance curves of monkeys across sessions.

We applied the cued block design to reduce monkeys’ confusion about the behavioral task. The cue appeared in the same location for at least 20 trials. To ensure that the cue directed the monkey’s spatial attention successfully, we applied the pre-experiment where the cue validity was 90% (*n* = 29). In the formal experiments, the color change only occurred at the cued position.

### 4.3. Electrophysiological Recordings

The micro-drive system with 32 electrodes (1.5 mm inter-channel spacing and 1.6 cm traveling length; Gray Matter Research, Bozeman, MT, USA) was used to obtain the electrophysiological data collected over 77 sessions (40 for monkey P, 37 for monkey S). As each electrode was separately regulated by a lead screw (125 μm/turn), we were able to change the depth of electrodes to obtain well-isolated neuronal activities.

### 4.4. Data Analysis

#### 4.4.1. Electrophysiological Data Analysis and Classification of Neurons

We performed spike sorting offline by Offline Sorter (Plexon Inc., Dallas, TX, USA). Data were band-pass-filtered at 300 to 4000 Hz and waveform segments were digitized at 40 kHz. We excluded the neurons whose waveform shape signal-to-noise (SNR) ratios were less than 2.4, amplitude SNR ratios were less than 1.2, and the percentage of short inter-spike interval (ISI < 1 ms) was over 0.2%. One hundred and sixty-five well-isolated neurons met the criteria above (monkey P: 80; monkey S: 85). One hundred and fifty-four neurons whose RFs were covered by gratings with neurons’ preference directions were paired and analyzed further. The average horizontal distances between two electrodes that recorded pairs simultaneously were 6.22 mm (range from 3.35 to 7.50 mm).

Recorded signals were extracted 500 ms time windows before the color change. Then, we calculated the spike counts in 10 ms bins and fitted them with the Gaussian function. Then, we calculated the response linearity (F1/F0) of the recording neurons. Simple cells produce a heavily modulated sinusoidal response to gratings (F1/F0 > 1) due to their segregated ON/OFF subregions within RFs. In contrast, complex cells feature overlapping ON/OFF subregions and generate a comparatively weak modulated response (F1/F0 < 1) [55].

#### 4.4.2. Receptive Field Mapping

Before starting the attention tasks, the location and size of the RFs were measured by the reversed subspace correlation method [56]. The RF centers of the recorded neurons remained stable across sessions (Appendix A). The average size of RFs across sessions (diameter: 1.11 ± 0.44°) was much smaller than the distance between the RF centers (2.41 ± 0.52°).

#### 4.4.3. Granger Causality Analysis

Multivariate autoregressive-based GC analysis treated signals as discretely sampled values of the continuous process, whereas spike trains were sequences of the point process. Thus, we measured directed causal communication between neurons’ spike trains by the likelihood-based point process framework GC analysis [17]. The likelihood functions of the spike trains of neuron *i* were calculated by the conditional intensity function (CIF). Then, we measured the causal relationship from neuron *j* to neuron *i* by calculating the reduction in the likelihood of spike trains of neuron *i* after excluding the spiking history of neuron *j*:(1)$$\Gamma_{ij}=log\frac{L_{i}^{j}}{L_{i}},$$
where $L_{i}$ was the likelihood of neuron *i* that contained all of the available covariates, while $L_{i}^{j}$ represented the likelihood of excluding the spiking history of neuron *j*.

Since the likelihood ratio was always less than or equal to 0, we further distinguished the excitatory and inhibitory influences of neuron *j* on neuron *i* by the sign of $\overset{M_{i}}{\underset{m=1}{\sum }}\gamma_{i,\;j,\;m}$ (average influence of the spiking history of neuron *j* on neuron *i*):(2)$$\Phi_{ij}=-sign\left(\overset{M_{i}}{\underset{m=1}{\sum }}\gamma_{i,\;j,\;m}\right)\Gamma_{ij},$$

The time interval was divided into *Mi* non-overlapping rectangular windows. $\gamma_{i,\;j,\;m}$ was the influence of neuron *j* on neuron *i* at the mth time window. A positive outcome indicates that neuron *j* has an excitatory impact on neuron *i*, while a negative result represents an inhibitory impact.

#### 4.4.4. Attentional Ratios and Multiplicative Attentional Ratios

We applied the widely used index to quantify the magnitude of spatial attentional effects on neuronal firing rates. The attentional ratios (*AR*) on firing rate (*FR*) were calculated as:(3)$$AR=\frac{FR_{IN}-FR_{OUT}}{FR_{IN}+FR_{OUT}},$$

The *AR* was calculated under AT conditions in this study, where $FR_{IN}$ represented the average firing rates when the attentional focus was inside the RFs. $FR_{OUT}$ were the average firing rates when the attentional focus was directed to the RFs of other neurons in pairs.

We then calculated the multiplicative attentional ratios (MAR) between neuron *i* and neuron *j* by multiplying the AR of each neuron.

#### 4.4.5. Laminar Alignment

As there were no multiple laminar probes on a single electrode in the electrophysiological recording array, we cannot use the current source density (CSD) methods to precisely identify the reference coordinate. Thus, the reference coordinate in our study was defined as the minimum depth at which electrodes can record the neurons’ activities. Based on the recording electrodes’ distance from the reference coordinate, neurons were divided into the three main laminar compartments: supra-granular (0–800 μm below the reference coordinate), granular (800–1200 μm below the reference coordinate), and infra-granular (1200–2000 μm below the reference coordinate). Appendix A shows the depth distribution of recorded neurons.

### 4.5. Statistical Analysis

We treated the session averages as independent samples to avoid underestimating confidence intervals and inflating false positive rates. Paired *t*-tests were applied to examine differences in detection accuracy between valid and invalid cues. For reaction time and detection accuracy, we used repeated-measures ANOVA (rmANOVA) involving cues’ positions as within-subject factors. For electrophysiological data, two-factor rmANOVA was used to test the changes in Granger causality. The contrast between various conditions was evaluated by using the Bonferroni-corrected post hoc analysis. At *p* < 0.05, statistical comparisons were considered significant.

## 5. Conclusions

Overall, we found spatial attention enhanced inhibitory connections between pairs with heterogenous visual input, which solely occurred in the S-C hierarchy within V1. Such an attentional-related enhancement is displayed significantly in the feedforward connection from simple to complex cells. Furthermore, information flowed more from simple to complex cells when attentional focus covered one of the paired neurons’ RFs. By exploring the effects of attentional focus placement, we found spatial attention did not initiate distinct information flow but rather amplified the directed communication in the S-C hierarchy. Furthermore, we found the attentional enhancement on hierarchical communications depended on the attentional modulation of neurons’ firing rates. Our findings enrich the hierarchical connection model of simple complex cells and help to better understand how spatial attention influences neuronal interactions within V1.

## Figures and Tables

**Figure 1 ijms-24-08229-f001:**
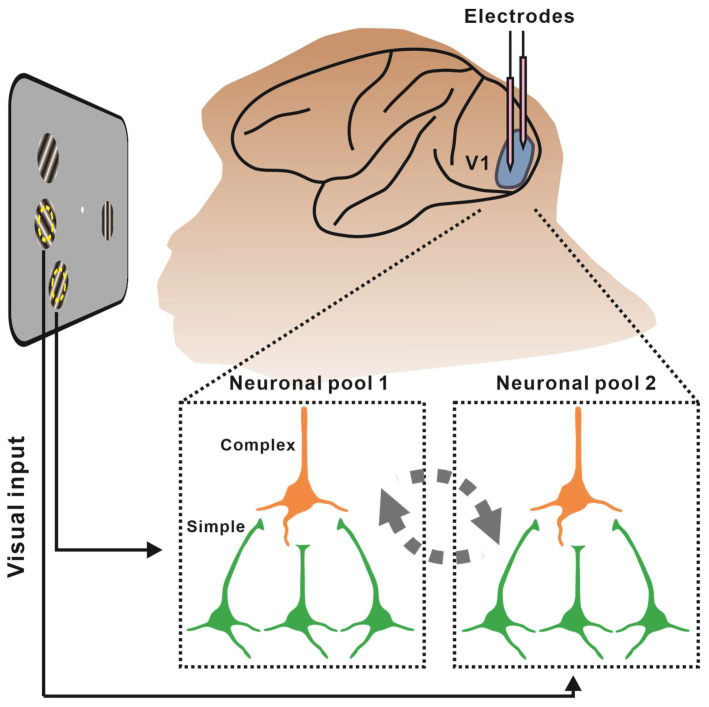
Study schematic. We simultaneously recorded neuronal pairs from V1 while monkeys performed a spatial attention task. The pairs we recorded had non-overlapping RFs and heterogeneous visual inputs. The neurons in yellow are complex cells, whereas those in green represent simple cells.

**Figure 2 ijms-24-08229-f002:**
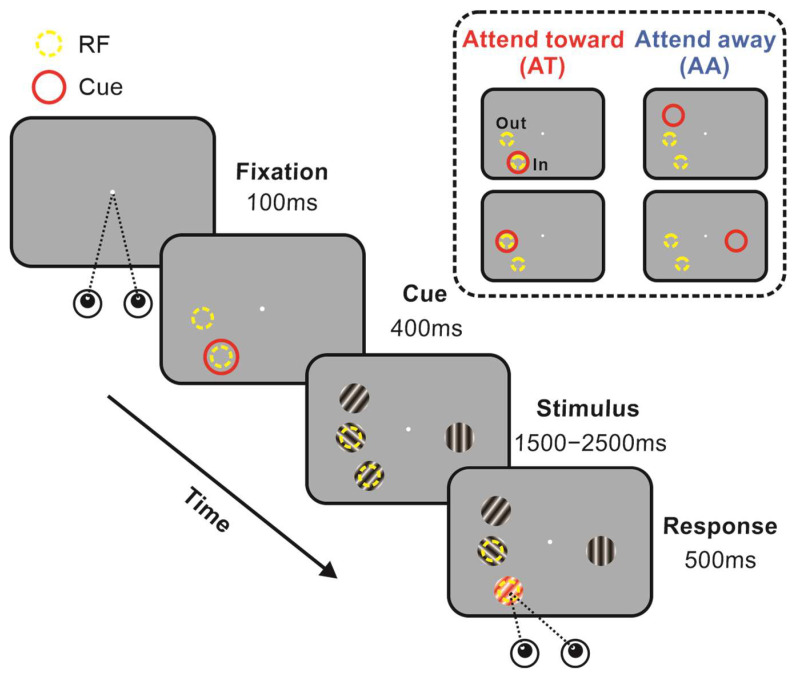
Illustration of behavioral task (**left**) and attentional conditions (**upper right**). Following fixation on the central spot, a red attention-directing cue appeared peripherally. Then four gratings were presented equidistant to the central spot: two stimuli covered pairs’ RFs, and the other two were located outside the RF. The color of the stimulus changed at unpredictable times. Monkeys were rewarded for making a successful saccade to that grating within 500 ms. The conditions in which the red ring encircled one of the neurons’ RF were defined as attend toward (AT), while the other conditions were defined as attend away (AA). The AT conditions were further divided into AT_In and AT_Out conditions according to the relative position of neurons’ RFs and cues.

**Figure 3 ijms-24-08229-f003:**
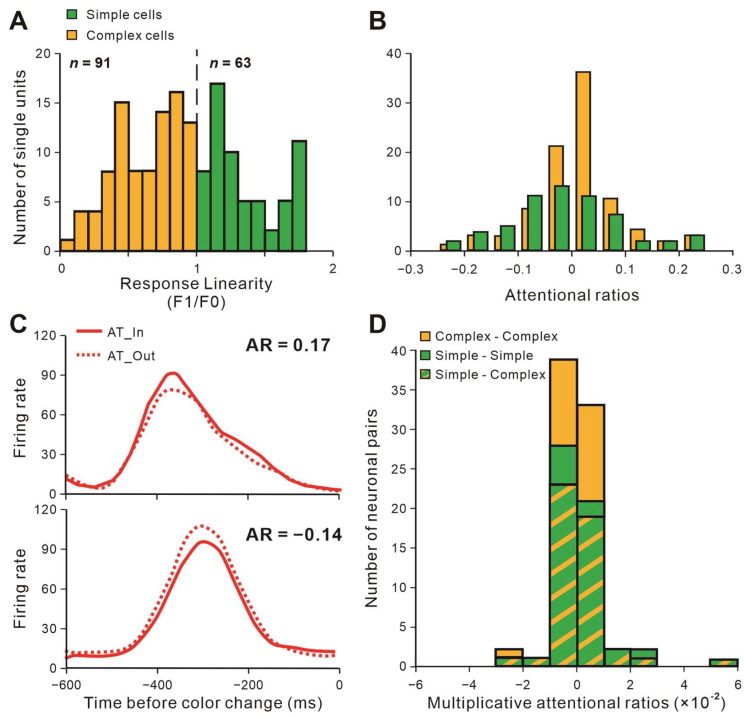
The characteristics of recorded neurons and examples of activities in V1. (**A**) The response linearity distribution of recorded neurons. (**B**) The distribution of simple and complex cells with varied attentional ratios. Attentional ratio = (AT_In − AT_Out)/(AT_In + AT_Out). (**C**) Neurons whose firing rates were influenced by attention. One of the neurons (upper) showed attention-related response enhancement (AR > 0), whereas the other one (down) showed suppression (AR < 0). (**D**) The number of pairs with different multiplicative attentional ratios among C-C, S-S, and S-C pairs.

**Figure 4 ijms-24-08229-f004:**
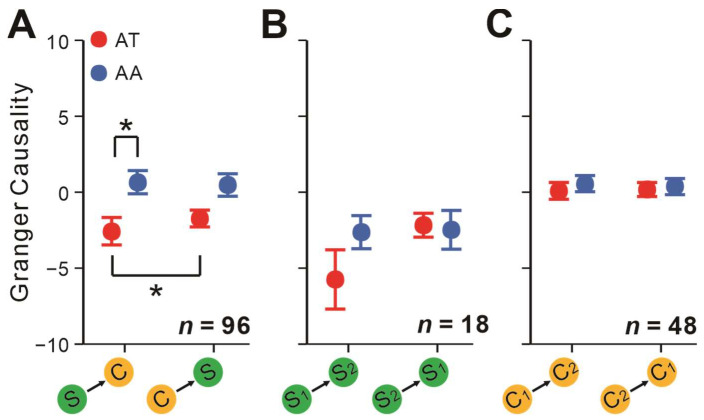
Attentional modulation of pairwise GCs among different groups of pairs. (**A**) GCs between simple and complex cells in AT (red) and AA conditions (blue). (**B**,**C**) Same as (**A**) but analyzed on S-S pairs and C-C pairs separately. Neurons from neuronal pool 1 were marked as S_1_ or C_1_, while those from neuronal pool 2 were labeled as S_2_ or C_2_ (neuronal pools 1 and 2 are shown in Figure 1). Repeated-measures ANOVAs and Bonferroni-corrected post hoc analysis, * *p* < 0.05. Error bars represent ± SEM.

**Figure 5 ijms-24-08229-f005:**
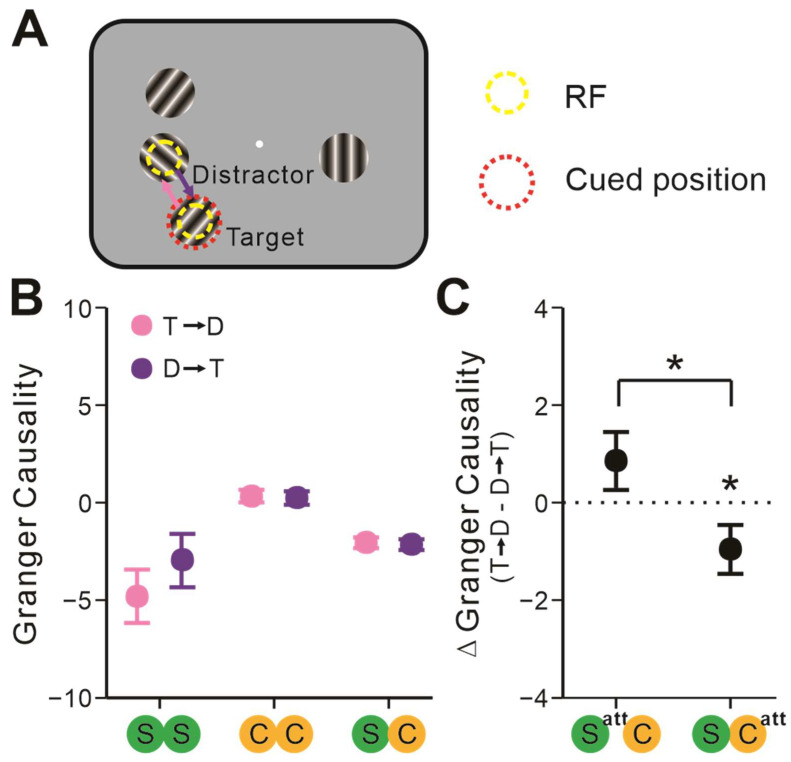
Attentional focus conditionally influenced neuronal interactions. (**A**) Illustrations of experimental conditions. The dashed red ring enclosed the target stimuli and attentional focus was directed to it by cue. The communicating directions from target-stimuli-covered neurons to distractor-stimuli-covered neurons (T → D) were highlighted in pink, and the opposite directions (D → T) were highlighted in purple. (**B**) The GC of these two directions among S-S, C-C, and S-C pairs. (**C**) ΔGC [(T → D) − (D → T)] among S-C pairs when the attentional focus covered simple or complex cells’ RFs. Repeated-measures ANOVAs and Bonferroni-corrected post hoc analysis, * *p* < 0.05. Error bars represent ± SEM.

**Figure 6 ijms-24-08229-f006:**
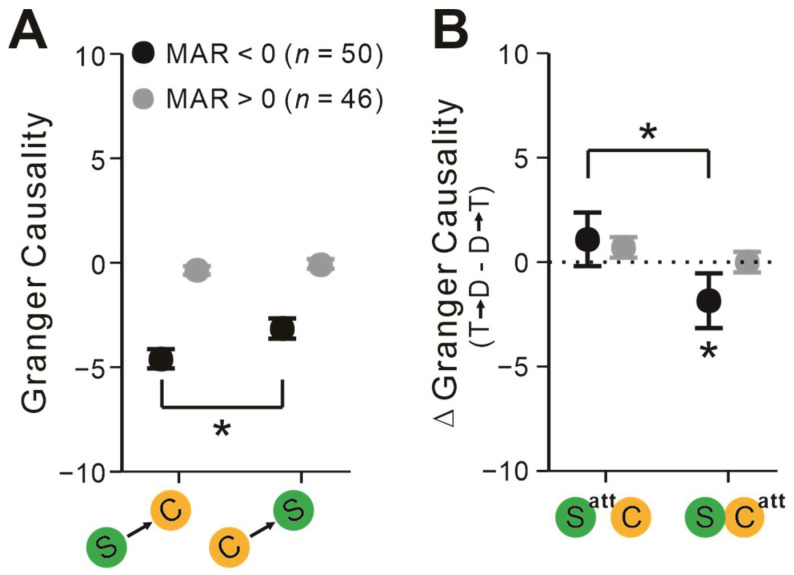
Effects of pairs’ MAR on neuronal interactions between simple and complex cells. (**A**) GCs between simple and complex cells when MAR was negative (dark) or positive (gray). (**B**) ΔGC among S-C pairs when MAR was negative or positive. Paired *t*-test, repeated-measures ANOVAs, and Bonferroni-corrected post hoc analysis, * *p* < 0.05. Error bars represent ± SEM.

**Figure 7 ijms-24-08229-f007:**
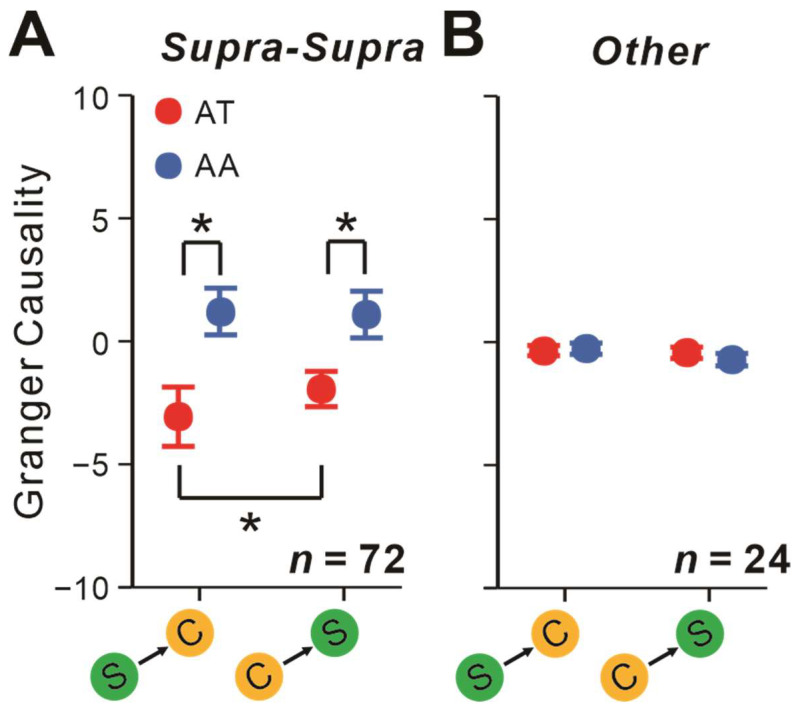
Effects of spatial attention on S-C pairs’ interactions between cortical layers. (**A**) GCs of pairs within supra-granular layers. (**B**) GCs of pairs where not all originated from supra-granular layers. Repeated-measures ANOVAs and Bonferroni-corrected post hoc analysis, * *p* < 0.05. Error bars represent ± SEM.

## Data Availability

The raw data supporting the conclusions of this study are available on request from the corresponding author.

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
