# Peer review of "Spatial Attention Modulates Neuronal Interactions between Simple and Complex Cells in V1"

_ijms, 2023, doi:10.3390/ijms24098229_

Round 1
Reviewer 1 Report
The current study examined whether and how spatial attention affects neuronal communication within the primary visual cortex, particularly between neural pairs that receive diverse visual input. To accomplish this goal, the researchers monitored the pairs’ activity of macaque monkeys while performing a spatial-attention-involved task and then used likelihood-based Granger causality analysis to investigate attentional modulation of neural interactions. The findings demonstrated a substantial attention-related drop in Granger causality in S-C pairs, most notably in the S-to-C feedforward connection. Second, the feedforward connection's interaction intensity was much larger than that of the feedback under attend toward conditions. The researchers came to the conclusion that spatial attention does not cause particular information flow but rather amplifies directed communication within the primary visual cortex.
The topic is interesting. Indeed, it is important to better understand how spatial attention influences neuronal interactions within primary visual cortex. The study is original and provides important results. Figures are useful. The article include relevant references in the field.
I would like to make some minor suggestions:
-It woud be better to clarify from the beginning the type of the study.
- The manuscript needs to be better organised to help reader follow. The structure and the analysis can be improved to better highlight the hierarchy of the article as well as the important points.
-The description of the methods as well as of the results can be improved.
- Please check that all abbreviated phrases should be written in full the first time that they are used.
Author Response
We thank this reviewer for emphasizing the significance of the study and for making suggestions on improving the manuscript, which is now fully addressed in this resubmission.
Point 1: It would be better to clarify from the beginning the type of the study.
Response 1: Thanks for your helpful advice. We submitted a hypothesis-based research article. We framed our research questions and the assumption by combing previous publications, then designed an experiment to test the hypothesis. We hope we have answered your questions correctly.
We have included the following hypothesis and conclusion in our revised manuscript:
Purpose and Hypothesis: Our goal was to determine whether and how spatial attention modulated communications between neuronal pairs in V1 with heterogeneous visual input. We proposed that spatial attention would enhance communication within the S-C hierarchy, following the findings reported in cross-visual-area studies. Moreover, pairs’ communication would be inhibitory due to heterogeneous visual input.
Conclusion: Our observation of significant attentional enhancement of inhibitory communication in the S-C hierarchy supports the hypothesis we came up with. By exploring the effects of attentional focus placement, we found spatial attention did not initiate distinct information flow but rather amplified the directed communication in the S-C hierarchy. Further, we found the attentional enhancement on hierarchical communications depended on the attentional modulation of neurons’ firing rates. Our findings enrich the hierarchical connection model of simple and complex cells and help to better understand how spatial attention influences neuronal interactions within V1.
Point 2: The manuscript needs to be better organized to help the reader follow. The structure and the analysis can be improved to better highlight the hierarchy of the article as well as the important points. The description of the methods as well as of the results can be improved.
Response 2: Thank you for your valuable advice. We reorganized our manuscript’s structure. To help readers follow, we briefly introduced the overall task, the manipulations of spatial attention, followed by definitions of all the variables (simple cells, complex cells, AR, and MAR) at the beginning of the result section. Referring to the requirements of the journal, details of the behavioral tasks and electrophysiological recording, as well as methods of data and statistical analysis, were mentioned in the Materials and Methods section. We also added more subheadings, subparagraphs, and summaries to assist readers in comprehending the important points of the article. Further, we recalibrated our statements in methods and results to reduce fuzzy and duplicate descriptions.
Point 3: Please check that all abbreviated phrases should be written in full the first time that they are used.
Response 3: We apologized for these mistakes. We took a closer look at our manuscript and made sure that all abbreviations were explained the first time they are used. Please check out the changes we made below:
AT (line 20, in the abstract) – attended toward (AT)
RFs (line 23, in the abstract) – receptive fields (RFs)
LGN (line 42, in the introduction) – lateral geniculate nucleus (LGN)
FEF (line 43, in the introduction) – frontal eye field (FEF)
MT (line 43, in the introduction) - middle temporal (MT)
SC (line 43, in the introduction) - superior colliculus (SC)
Reviewer 2 Report
Dear Editor and Authors,
I read the paper entitled ‘Spatial Attention Modulates Neuronal Interactions between Simple and Complex Cells in V1' with great interest.
This paper investgated whether and how spatial attention modulates neuronal communication within V1, especially for neuronal pairs with heterogeneous visual input. Authors simultaneously recorded the pairs’ activity from macaque monkeys when they performed a spatial-attention-involved task, then applied likelihood-based Granger causality analysis to explore attentional modulation of neuronal interactions.
The article covers the topic exhaustively.
The title describes the core message of the paper.
The abstract incorporates key messages, in a concise manner.
The structure of the paper is accurate.
Authors conclude that spatial attention does not induce specific information flow, but rather amplifies directed communication within V1.
Author Response
We extend our sincere gratitude for all the hard work you put into reviewing our manuscript, and appreciate your affirmation of our study.
In this revised manuscript, we stated our purpose and hypothesis at the end of the introduction and concluded by confirming our hypothesis. We also reorganized our manuscript’s structure and added more subheadings to assist readers in comprehending the important points of the article. The limitations of this study were added to the discussion.
Reviewer 3 Report
Dear Authors,
the presented manuscript is interesting. In my opinion, it should be adapted to the requirements of the Journal. The results section is clearly presented. However, the purpose of the work should be clearly defined in advance. The discussion is too poor. Please present the results against the background of other studies. The summary section and the limitations of the study section are missing, because there were certainly some. Please arrange the literature according to the requirements of the journal.
Author Response
We thank this reviewer for emphasizing the significance of the study and for making suggestions on adjusting the manuscript, which is now fully addressed in this resubmission.
Point 1: The purpose of the work should be clearly defined in advance.
Response 1: Thank you for your suggestions. The purpose of our work has been stated in the last paragraph of the introduction section. The modifications are as follows:
Our goal was to determine whether and how spatial attention modulated communications between neuronal pairs in V1 with heterogeneous visual input. We proposed that spatial attention would enhance communication within the S-C hierarchy, following the findings reported in cross-visual-area studies. Moreover, pairs’ communication would be inhibitory due to heterogeneous visual input.
Point 2: The discussion is poor. Please present the results against the background of other studies. The summary section and the limitations of the study section are missing, because there were certainly some. Please arrange the literature according to the requirements of the journal.
Response 2: Thank you for your valuable advice, we have improved our discussion in the revised manuscript. First, we concluded by confirming our hypothesis: “Our observation of significant attentional enhancement of inhibitory communication in the S-C hierarchy supports the hypothesis we came up with.” Second, the comparison between our results and other studies has been integrated into a separate paragraph: “Our findings are also consistent with previous trans-cortical and homo-cortical studies, which demonstrated attentional-related facilitation in neuronal communications. However, the underlying mechanism in our study may differ from these studies. In these previous studies, neurons’ RFs were covered by a sufficiently big stimulus, which means they would cooperate to recognize the target. But neurons in our study received different visual inputs, implying that the pattern of communication was different. That is, when the attentional focus switched away from neurons’ RFs, pairs performed a similar interference function in identifying targets. In this case, the communication between them is meaningless and energy-consuming. But when the attentional focus shifted to one neuron’s RF, the information from the other neuron became interference. Based on that, spatial attention would suppress the noise to improve the efficacy of encoding sensory.” Finally, the limitations in our study were raised by comparing it with other studies: “First, most neurons recorded in our study were obtained from supra-granular layers to ensure that the effect of attention on neuronal activities was strong enough. Follow-up studies could use multichannel linear electrodes to record neuronal activities from multiple layers simultaneously. By doing so, we can better understand the hierarchical structure of simple and complex cells in V1 and the role of attention in the early stage of visual processing. Second, each electrode in the recording system we used can be controlled independently by a screw. But compared to other electrophysiological recording systems (like the Utah array), the channel spacing is large and the number of neurons that can be recorded in each session is limited. It may be advantageous to utilize the Utah array to simultaneously record the activities of pairs that received homogeneous or heterogeneous visual input.” The references in the manuscript have been cut following the requirements of the journal.
Reviewer 4 Report
This article reports experimental investigation of an important issue that is both theoretically interesting and practically significant: impact of attention on the dynamics of neuronal interaction. The experimental setup and selected analytic techniques reflect the state-of-the-art and are appropriate for the task at hand. The results are consistent with those in the earlier studies but are still likely to be of interest to the audience intended by the journal.
Regrettably, the style of presentation is somewhat redundant and convoluted, making it difficult to appreciate the findings. This reviewer strongly recommends re-arranging components of the paper in a linear fashion starting with the overall task and definitions of all the variables (simple cells, complex cells, etc.), followed by introducing all the parameters that will be recorded and/or manipulated, followed by describing the setup designed to implement these manipulations, methods of data analysis, and so on, down to describing the findings and their interpretations. In the present text, these components are scattered and partially repeated throughout the text.
It would be helpful to start with a hypothesis motivating the study (e.g., attention can be expected to enhance both feedforward and feedback interactions in the S-C hierarchy) and conclude with confirming or disconfirming the hypothesis. Passages that are fuzzy and speculative (e.g. lines 214-218) should be explained or, preferably, excluded.
English is good but some editing is required.
Author Response
We thank this reviewer for emphasizing the significance of the study (‘…both theoretically interesting and practically significant…’) and for raising several important issues in describing the findings, which are now fully addressed in this resubmission.
Point 1: The style of presentation is somewhat redundant and convoluted, making it difficult to appreciate the findings. Components of paper are scattered and partially repeated.
Response 1: Thank you so much for your sincere advice, we have adjusted the structure of our manuscript and streamlined some duplicate components.
Referring to the prescribed format guidelines for journals, we provided sufficient details about the methods and materials at the end of the paper. To help readers follow, at the beginning of the results section, we briefly introduced the overall task, the manipulations of spatial attention, followed by definitions of all the parameters (simple cells, complex cells, AR, and MAR). Details of the behavioral tasks and electrophysiological recording, as well as methods of data and statistical analysis, were mentioned in the Materials and Methods section.
Furthermore, we reduced repetitive statements of results in both the introduction and discussion sections. We put the summary of the findings in the conclusion section.
Point 2: It would be helpful to start with a hypothesis motivating the study (e.g., attention can be expected to enhance both feedforward and feedback interactions in the S-C hierarchy) and conclude with confirming or disconfirming the hypothesis.
Response 2: Thanks for your valuable comments. We took your suggestions and included hypothesis and conclusions in our manuscript. The modifications are as follows:
Hypothesis: Our goal was to determine whether and how spatial attention modulated communications between neuronal pairs in V1 with heterogeneous visual input. We proposed that spatial attention would enhance communication within the S-C hierarchy, following the findings reported in cross-visual-area studies. Moreover, pairs’ communication would be inhibitory due to heterogeneous visual input.
Conclusion: Our observation of significant attentional enhancement of inhibitory communication in the S-C hierarchy supports the hypothesis we came up with.
Point 3: Passages that are fuzzy and speculative (e.g. lines 214-218) should be explained or, preferably, excluded.
Response 3: Sorry for the fuzzy statement. To respond to the hypothesis in the introduction section and reduce repetitive statements of results, we have modified this part (lines 214-218) as “Our observation of significant attentional enhancement of inhibitory communication in the S-C hierarchy supports the hypothesis we came up with. Subsequently, we would further compare our findings with previous publications, elucidate the physiological implications of the results, and point out the limitations and prospects for potential future explorations.”
These fuzzy passages have been clarified in the conclusion sections: “By exploring the effects of attentional focus placement, we found spatial attention did not initiate distinct information flow but rather amplified the directed communication in the S-C hierarchy. Further, we found the attentional enhancement on hierarchical communications depended on the attentional modulation of neurons’ firing rates.”
Point 4: English is good but some editing is required
Response 4: Thanks for your reminder. We have gone over our manuscript carefully and made adjustments (e.g., use more connectors (i.e., furthermore, hence, in contrast, then…) instead of “and” at the beginning of a sentence).
Reviewer 5 Report
Authors presented an elegant study on how spatial attention influences visual cortex neurons activity in monkeys. The information is presented clearly and in a reproducible way. I believe that researchers interested in the field may take advantage from the presented research.
I have just few minor concern that may slightly improve the manuscript:
- in the introduction, authors mention their results (line 66); the introduction should only contain a background with the rationale of the study and the objectives of the research. If they are referring to previous works of the same group, then mention them with a reference.
- There are abbreviations in the abstract never explained before (e.g., AT line 20 and RFs line 23); same in the introduction (line 42), while simple and complex visual cortex neurons had been shortened during the abstract, but not when they first appear in the main text (line 43).
Quality of English is quite fine, it needs just few adjustments: e.g., during the text, authors often begin a sentence with "and". Try to use more connectors.
Author Response
We thank this reviewer for emphasizing the significance of the study (‘Authors presented an elegant study on how spatial attention influences visual cortex neuronal activity in monkeys.’) and for helping us adjust our manuscript.
Point 1: In the introduction, authors mention their results (line 66); the introduction should only contain a background with the rationale of the study and the objectives of the research. If they are referring to previous works of the same group, then mention them with a reference.
Response 1: Thanks for your professional advice. The results mentioned in the introduction (starting from line 66) have been removed. We have further stated the objectives and hypothesis in the introduction sections. The modifications are as follows:
Our goal was to determine whether and how spatial attention modulated communications between neuronal pairs in V1 with heterogeneous visual input. We proposed that spatial attention would enhance communication within the S-C hierarchy, following the findings reported in cross-visual-area studies. Moreover, pairs’ communication would be inhibitory due to heterogeneous visual input. Toward the object, we first explored whether there is an attentional enhancement of communication in V1 or sorely within the S-C hierarchy. We then examined how spatial attention modulated the communications within this hierarchy, as well as the key factors that influenced the attentional modulation of hierarchical communications.
Point 2: There are abbreviations in the abstract never explained before (e.g., AT line 20 and RFs line 23); same in the introduction (line 42), while simple and complex visual cortex neurons had been shortened during the abstract, but not when they first appear in the main text (line 43).
Response 2: We apologized for not explaining abbreviations when they first appeared. We have clarified all the abbreviations in both the abstract and introduction sections. The detailed explanations are as follows:
AT (line 20) – attended toward (AT)
RFs (line 23) – receptive fields (RFs)
LGN (line 42) – lateral geniculate nucleus (LGN)
FEF (line 43) – frontal eye field (FEF)
MT (line 43) - middle temporal (MT)
SC (line 43) - superior colliculus (SC)
Point 3: During the text, authors often begin a sentence with "and". Try to use more connectors.
Response 3: Thanks for your valuable suggestions. We have taken them on board and made some adjustments by using more connectors (i.e., furthermore, hence, in contrast, then…) instead of “and”.
Round 2
Reviewer 3 Report
Manuscript is acceptable in present form.
Reviewer 4 Report
The comprehensiveness of the discussion and the quality of the presentation have been substantially improved.